# How Neighbors Influence Rice–Crayfish Integrated System Adoption: Evidence from 980 Farmers in the Lower and Middle Reaches of the Yangtze River

**DOI:** 10.3390/ijerph20054399

**Published:** 2023-03-01

**Authors:** Ke Liu, Zhenhong Qi, Li Tan, Canwei Hu

**Affiliations:** 1School of Management, Wuhan Polytechnic University, Wuhan 430048, China; 2College of Economics and Management, Huazhong Agricultural University, Wuhan 430070, China

**Keywords:** neighborhood effect, rice–crayfish integrated system, technology adoption

## Abstract

Rice-aquatic animal integrated systems can alleviate food and environmental insecurity. Understanding how this practice is adopted by farmers is significant for promoting the development of the agricultural industry. Given the information inadequacy and information frictions in agricultural society in China, farmers are susceptible to the behaviors of their neighbors through social interaction. This paper defines neighboring groups that are both spatially and socially connected to identify whether neighbors influence farmers’ adoption of rice–crayfish integrated systems using a sample in the lower and middle reaches of the Yangtze River in China. The findings reveal that for every one-unit increase in neighbors’ adoption behavior, the probability of farmers’ adoption increases by 0.367 units. Therefore, our results may have great value for policymakers seeking to take advantage of the neighborhood effect to complement formal extension systems and promote the developments of China’s ecological agriculture.

## 1. Introduction

In recent years, rice–aquatic animal integrated systems (i.e., co-farming with aquatic animals such as crayfish, crab, soft-shelled turtle, etc.) have gained increasing attention for their potential for alleviating food and environmental insecurity. Rice–aquatic animal integrated systems can bring about high yields and low environmental impacts and have been widely explored through field experiments [1,2] and household surveys [3]. Among them, rice–crayfish integrated systems have experienced explosive growth in China since 2016 and are considered to be a valid approach for ensuring the supply of food and aquatic products, increasing farmers’ incomes and promoting rural revitalization [4,5]. Rice–crayfish integrated systems allow for the efficient internal recycling of crayfish and rice. In this practice, on the one hand, the rice field provides a habitat for crayfish, and the straw in the field creates a heat preservation effect which facilitates the hatching of crayfish seedlings. Meanwhile, rice straw corrosion facilitates the growth of plankton in the water, which are regarded as nourishment for crayfish and effectively address the straw burning issue in China. On the other hand, integrated farming can take advantage of agricultural byproducts to decrease dependence on agroindustry inputs such as fertilizers and pesticides. To be more specific, crayfish digest and utilize rice straw and can eliminate the presence of pests in the straw. The excreta of crayfish also supply organic fertilizer for rice growth, and the crayfish in paddy fields constrain the use of pesticides and fertilizers due to their being sensitive to chemical inputs. Thus, integrated systems are regarded as an ecological agricultural practice.

According to the statistical data, rice–crayfish integrated systems constitute 52.95% of the total land areas used for rice–fish integrated systems. Additionally, 83.54% of the total production of crayfish is through rice–crayfish integrated systems [6]. The increased use of this type of integrated system can be partly attributed to political subsidies, field demonstrations and technical instructions from local extension agents. However, the implementation of rice–crayfish integrated systems involves intensive knowledge and requires deep insight into recycling and execution to ensure high crayfish and rice yields and a low environmental impact. Thus, farmers are not able to master the core techniques through simple learning and typical demonstration visits. Moreover, integrated systems require large investments (e.g., proprietary equipment) and involve increased flood and drought associated risks [7]. Clarifying the micro-mechanisms of farmers’ adoption of rice–crayfish integrated systems is significant for promoting the development of ecological agricultural practices in general.

In Chinese culture, which emphasizes collectivism and “acquaintance”, neighbors are considered to be one essential driver of family decision making [8]. Given the information inadequacy in agricultural society in China and the existing information frictions between farmers and extension agents, farmers share farming information and techniques with their neighbors and assist each other with farm work. Through these frequent and close interactions, farmers are influenced by the behaviors of their neighbors [9], which forms a neighborhood effect. Studies have demonstrated that farmers in proximity to each other tend to have similar adoption behavior toward new technologies to reduce learning costs through information sharing [10]. Even though the role of the neighborhood effect is acknowledged in the literature, many studies focus on geographical criteria [11,12] (i.e., distance and location) to define neighboring units and have not specified the strength of interactions between neighboring units. The presence of the neighborhood effect usually amplifies this effect, which implies that a multiplier effect exists through social interaction that is strongly conditioned by the geographic distance between individuals [13]. Moreover, many studies have examined the neighborhood effect on simple technology adoption (e.g., biogas adoption [14] and water conservation [15]).

Our study expands the literature in two aspects. First, we define neighboring groups at the village level using samples of small communities. Individual farmers and their neighbors are both spatially and socially connected, which indicates the existence of real— and thus more relevant—social interactions. Usually, community-based definitions of neighbors or peers are too broad and may include irrelevant reference individuals. However, this concern may not appear in our village-based sample for the following reasons. Families in one village usually have lived there for generations and the farmers are familiar with each other, which maintains strong social relationships between farming households [16]. The natural and exogenous characteristics of rural villages suggest that the definition of neighbors in our sample includes both friends and non-friend acquaintances and excludes strangers. Such village-based neighboring groups are not self-selected networks; thus, their exogeneity is unlikely to interfere with our desired outcome. Second, little empirical research has concentrated on exploring the relationship between the neighborhood effect and farmers’ integrated farming system adoption. Our paper expands the existing literature by providing direct evidence of the neighborhood effect in integrated farming system adoption.

In this article, we consider both geographic and socioeconomic criteria when defining the neighbor group to examine the presence of the neighborhood effect in farmers’ rice–crayfish integrated system adoption behavior. Understanding how new practices are disseminated through these interactions is helpful for developing agricultural policies that target specific agricultural areas or communities or even farmers where certain technologies should be introduced to achieve the desired impact.

The reminder of this paper is structured as follows. Section 2 is a review of the literature. Section 3 describes the data and empirical methods. In Section 4, we present the results and a discussion. Section 5 concludes.

## 2. Literature Review

Previous studies have identified many factors of farmers’ adoption behavior and indicated that subsidies, agricultural extension services, field schools, and field demonstrations can improve farmers’ adoption rates [17,18]. Studies have also suggested that household characteristics, such as age, education, farm size, and income level as well as perception of technology adoption, environmental concern, behavioral goals, and attitude have important roles [19,20,21,22,23]. The costs and benefits of the technology also affect farmers’ adoption decisions [24,25]. Furthermore, the nudge theory, or “altering people’s behavior in a predictable way without forbidding any options or significantly changing their economic incentives,” has been widely discussed as a factor in farmers’ adoption behavior [26,27]. In addition to these factors, economists and policymakers have argued that individual behaviors vary with the behavior of the group through mechanisms other than economic aspects [15,28,29], namely, the neighborhood effect.

Many studies have confirmed the presence of the neighborhood effect in farmers’ decision-making processes in settings ranging from rural housing demand [30], rural labor mobility [31], commercial health insurance purchasing [32], and response to climate change [33]. Exploring the impact that neighbors or social interactions have on individual farmers’ technology adoption have also been widely explored. One strand of the literature focuses on the mechanism of the neighborhood effect in terms of information sharing and social norms. Xiong and Payne [34] investigated how peer effects occur and found that family members sharing experimental resources and production externalities between contiguous plots of land positively impacts farmers’ Artemisia slengensis (AS) adoption. Di Falco and Doku [35] argued that the peer effect occurs through information diffusion by observing peer farmers’ choices, which encourages farmers to adopt multiple climate adaption strategies at the household level. Tran-Nam and Tiet [36] considered organic farming neighbors or peers as a source of information, knowledge, and motivation to help farmers transition to organic farming. Crudeli and Mancinelli [37] focused on peer approval and examined how the social norm of being a “good farmer” influences farmers’ innovation adoption.

Another strand of research has concentrated on identifying the presence of the neighborhood effect. Many studies have identified the spatial or geographic neighboring effect. Sampson and Perry [11] take spatial bands around each water right as a peer group and find that spatial neighboring effects in the adoption of LEPA (i.e., low-energy precise application) diminishes with distance. Bollinger and Burkhardt [15] found peer effects in water conservation. Their identification strategy relies on quasi-experimental variation from consumer migration in which new households move into peer groups and make water consumption and landscape changes. Kolady and Zhang [38] use location-specific survey data to define farmers’ peer group through physical proximity, and the results show that spatially mediated peer effects are important in the adoption of conservation tillage and diverse crop rotation. Skevas and Skevas [20] discovered that peer effects arise from both nearby farmers’ adoption of unmanned aerial vehicles and the spatial spillover of other farmers’ characteristics.

More research has identified the neighborhood effect through social interactions between individuals and their neighboring group. Gao and Grebitus [39] reveal that hog farmers’ genomics adoption time frames are positively correlated with other closely related hog farmers’ time frames. Ward and Pede [40] define same-village membership and geographical distance as spatial network systems and demonstrate that the distance between hybrid rice adopters affects farmers’ adoption of hybrid rice.

## 3. Data and Methods

### 3.1. Data

The data of this study are from a survey of rice farmers we conducted in the provinces of Hubei, Hunan, and Anhui in the middle and lower reaches of the Yangtze River of China in July 2019. These three provinces are the main originating location of rice–crayfish integrated systems in China. This region is characterized as a subtropical monsoon with a humid climate and an average annual temperature ranging between 14 °C and 18 °C and a forest-free period ranging between 210 and 270 days. The annual average precipitation is approximately 1000–1500 mm. These three provinces were chosen as the study area for the following reasons. First, they are major producers of rice and aquaculture products—especially crayfish—in China due to their climate conditions and the rich water resources in the middle and lower Yangtze Plain. Rice–crayfish integrated systems were first developed in Jianli, Hubei province, and over the years, this cultivation system has been adopted by a growing number of farmers in this region. Second, local governments in this area recognize the environmental and economic benefits of such systems and thus promote them using a wide range of policy instruments, from offering direct subsidies to farmers who adopt them to providing technical assistance through agricultural extension services. As a result, it is estimated that these three provinces have a great amount of farming areas that use rice–crayfish integrated systems, among which Hubei province ranks first [6]. A map of the study area is shown in Figure 1.

A multistage stratified sampling procedure was used to choose a representative sample of rice farming households in the region. In the first stage, three counties within each province were chosen to account for the distribution of land used for rice–crayfish integrated systems and the general level of economic development within these provinces. In the second stage, around 1000 rice farming households were randomly chosen from villages in each county. Most farming families have lived in their village for generations and are within walking distance of each other. We used a structured questionnaire to obtain farmers’ information. The questionnaire consisted of six parts: household and farm characteristics (e.g., age, education, farm size, labor, and assets); sources of new technology; crop planting methods (e.g., the adoption of rice–crayfish integrated systems) and related inputs and outputs; farmers’ utilization of agricultural socialization services (e.g., agricultural mechanization services); farmers’ perception of rice–crayfish integrated systems; and village characteristics (e.g., infrastructure). We conducted face-to-face interviews with farmers through trained qualified postgraduates majoring in agricultural economic management in our research group based on a survey questionnaire. Since this study analyzes the influence of group behaviors, we deleted samples (≤3) with fewer than three neighbors. After dropping observations with missing information for key variables, we obtained a final sample of 980 households, 695 of which had adopted rice–crayfish integrated systems to some degree.

### 3.2. Methodology and Variables

There are several situations that may have led us to mix other effects with the neighborhood effect when we observed similar behavioral outcomes between individual farmers and their neighboring group. Therefore, measuring the neighborhood effect presents several challenges [13,40,41,42,43] which include: (1) the contextual effect, which reflects the fact that neighbors’ exogenous characteristics will directly affect individuals’ behavior (i.e., a farmer’s propensity to adopt will be affected by the mean age within their neighboring group); (2) the correlation effect, which indicates that individuals behave similarly in one group with which they tend to have similar characteristics or are confronted with a common set of unobserved characteristics (i.e., farmers may be affected by regional policies, such as the same agricultural subsidy policy, to have the same behaviors); (3) the self-selection problem, which implies that individuals select neighbors based on their preferences and backgrounds and have similar behaviors simply resulting from similar income levels or proximity; and (4) the reflection problem, by which individuals and their neighboring group make decisions or behave simultaneously. As a result, individuals forming a unilateral causal relationship with their neighboring group will cause an endogeneity problem.

To overcome the above problems, we applied a set of empirical strategies. We collected samples of 980 farming households in three major provinces in the lower and middle reaches of the Yangtze River in China in 2019, and formed plausible empirical neighbor groups given the small community nature of rural villages. In the context of our research setting, there are typically strong socio-economic ties in Chinese culture and thus we define farmers living in the same administrative village as a neighboring group. The reason for this is that choosing a village as a dataset contributes to solving the self-selection issue to some extent [30]. On the one hand, the household registration system of China and restrictions on rural–urban mobility hinders migration to villages. On the other hand, the formation of a village often spans generations [43]. Households settled in rural areas are unlikely to choose their neighbors through migration [44,45]. Moreover, this paper took neighboring farmers’ background characteristics, village characteristics, and provincial dummy variables into account to limit the importance of contextual and correlation effects [43]. Last, to estimate the effect of neighboring farmers’ adoption, we applied the instrumental variable method (IVs) to overcome the simultaneity issue and identify two exogenous variables as instrumental variables, “Village diversity in surnames” and “The proportion of paddy field area in the village”, to improve our identification of the neighborhood effect. After that, we conducted several robustness checks to confirm the presence of the neighborhood effect.

Since the explained variable, farmers’ adoption behavior, is binary, we chose the probit model as a benchmark model [45,46]. The basic formula is specified as follows:(1)$$Probit\;\left(Adoption_{i}^{c}\;=1\right)=\varphi (\beta_{0}+\beta_{1}NAdoption_{-i}^{c}+\beta_{2}X_{i}+\beta_{3}Y_{-i}^{c}+\beta_{4}Z_{i}+ProvinceDummy.$$

In this formula, $Adoption_{i}^{c}$ is an indicator of the rice–crayfish integrated system adoption of farmer i in village c (1 = yes; 0 = no). These data stem from a question in the questionnaire, namely, “Does your family adopt a rice–crayfish integrated system?”

The key explanatory variable is $NAdoption_{-i}^{c}$ (i.e., the neighborhood effect), which indicates the average adoption within a neighboring group, except for farmer i. The size and significance of the coefficient on $\beta_{1}$ are of particular interest to us. To ensure the accuracy of the results, the scope of “neighborhood” must be cautiously defined. Thus, we calculated the neighborhood effect using the following equation:(2)$$NAdoption_{-i}^{c}=\frac{\sum_{1}^{n}Adoption_{i}^{c}-Adoption_{i}^{c}}{n-1}$$

Equation (2) denotes neighboring farmers’ behavior in this paper. Neighbors’ influence should exclude the effects of the focal farmer; thus, farmer i is not included. c is the number of sampled farmers in the village.

$X_{i}$ is a vector of exogenous characteristics of the sampled farming household, including the age of the head of the household, education, risk preference, job status, perception of the economic benefits of rice–crayfish integrated systems, agricultural extension training attendance, scale of operations, agricultural labors, investment, cooperation membership status, proportion of agricultural income to total household income, and the furthest distance between two plots.

$Y_{-i}^{c}$ denotes a vector of neighbors’ characteristic variables. To minimize the contextual effect, we controlled neighbors’ head of household age, education, job status and cooperation membership status in the basic regression. The calculations followed Equation (2) (i.e., the average value within the neighboring group, but not the focal farmer in the same village).

Two measures were taken to suppress the correlation effect issue: conducting a province-varying fixed effects model and controlling village-based variables ($Z_{i}$), including the proportion of the effective irrigated area in the village and the effective traffic rate of the village’s road. The details of the variables are presented in Table 1.

As mentioned above, endogenous threats that arise from simultaneity should be controlled [45,46,47]. We applied the IV method to control the reflection problem [29]. We followed Gaviria and Raphael [48], Li and Zang [45] and Ling and Zhang [49] and select two exogenous natural characteristic variables as instruments. It must be clarified that the IV variables were not related to the individual adoption probability of the focal farmer because these two variables were considered exogenous natural characteristics and did not significantly affect the adoption behavior of individual farmers. Second, they were related to the mean adoption behavior of the endogenous neighborhood farming group.

## 4. Empirical Results and Discussion

### 4.1. Baseline Results of the Neighborhood Effect on Farmers’ Adoption Behavior

In this study, we began by identifying the neighborhood effect in farmers’ rice–crayfish integrated system adoption behavior. The empirical results are shown in Table 2, which reports the probit model, fixed effect (FE), and instrumental variable (IV) estimates in Column 1, Column 3, and Column 5, respectively. To compare the coefficients, all results are reported as the marginal effect of the variables in all tables, and all specifications control the impact of household characteristics, neighboring farmers’ characteristics, and village characteristics.

First, in Models 1 and 2, it can be seen that the coefficients on the neighborhood effect are both positive and significant at the 1% level, as expected. The size and significance of the coefficients do not change much (i.e., from 0.426 to 0.379), which indicates that farmers’ probability of adoption increases by 0.379 percentage points for each percentage point increase in the neighbors’ adoption rate. The results of Models 1 and 2 preliminarily confirm that the average integrated system adoption behavior within neighboring groups have a significantly positive influence on farmers’ adoption behavior. Thus, the neighborhood effect exists in farmers’ rice–crayfish integrated systems adoption.

Our focused specification is Model 3 (i.e., the IV method). As Manski [28] points out, individuals and their reference group can affect each other simultaneously, which can cause an endogeneity problem. We applied the instrumental variable method to solve this problem. For instruments, we consider whether the village is diverse in surnames and the proportion of paddy field area to cultivated land. The maximum likelihood estimation (MLE) is used to acquire the marginal effect of the IV probit model. The results are shown in Model 3 in Table 2.

Column 5 in Table 2 presents the IV probit estimates. The results suggest that as neighboring groups’ adoption improves by one percentage point, and farmers’ likelihood of rice–crayfish integrated systems adoption increases by 0.367 percentage points. Compared to Models 1 and 2, the coefficient on neighborhood effect has a noticeable decrease. This result also proved that the probit and FE models both overestimate the neighborhood effect. Taken together, this IV probit estimation supports the hypothesis that changes in neighbors’ adoption behaviors will in turn affect the focal farmer’s adoption behavior. One possible explanation for this finding is that focal farmers presume that their neighbors possess superior information. It is also possible that certain farmers are afraid to become “special” under the cultural background of the “Doctrine of the Mean” in China, so they tend to behave like their neighbors. This finding is in accordance with those of Di Falco and Doku [35] and Tran-Nam and Tiet [36].

The first-stage estimation results in Panel B indicate that the first IV (“Village diversity in surnames”) negatively affects neighbors’ adoption behavior, and “The proportion of paddy field area to cultivated land in the village” positively affects it. The degree of communication and trust between the farming households in mixed villages is relatively low compared to that in non-mixed villages, and the mutual influence between the farmers is relatively small, which may decrease the adoption effect. Rice–crayfish integrated systems are suitable for production in flat and water-rich fields, and good natural conditions may increase farmers’ output and revenue. Therefore, the higher the proportion of paddy field area in the village, the higher the possibility that farmers in the village will adopt integrated systems.

In addition, we empirically examined the validity of the instrumental variables using a series of tests. To exclude the assumption of weak instrumental variables, we used the two-step method (2SLS) to report the first-stage estimates. As shown in Panel B in Table 2, the F-statistic is 61.23 with a *p*-value of less than 1% (0.000), which implies that the weak-instrument issue should not be a concern in our estimates. The Amemiya–Lee–Newey minimum *p*-value of the over-identification test is 0.632, which is higher than 0.1. This result indicates that the joint null hypothesis should not be rejected, and the over-identification restriction is satisfied. Additionally, the *p*-value of the Durbin-Wu-Hausman test is 0.090, which rejects the null hypothesis. This result proves that variable of the neighborhood effect (*NE*) is endogenous, which implies the existence of the endogeneity problem. Thus, the chosen variables are valid as the instrumentals for the neighborhood effect.

Moreover, many control variables have a significant effect on farmers’ adoption behavior (e.g., ‘information access’ is an indicator to measure a farmer’s openness). Farmers who have more information access are more likely to obtain pro-adoption information and be open-minded to produce market-oriented products. This conclusion has been suggested in previous studies, which find that accesses to extension services and peers predict technology adoption [50,51]. The “perception of economic benefits” and “perception of population” items reflect farmers’ perception and judgement of rice–crayfish integrated systems, both of which result in a higher probability of adoption behavior. Investment proportion indicates the farmer’s adoption capacity; the higher the proportion of their own capital investment to the whole agricultural investment, the less likely they are to adopt. A possible reason for this finding is that their self-owned funds are relatively sufficient, thus indicating that their economic situation is good. This proves that the original allocation efficiency of farmers’ funds, land, and labor is high. Therefore, farmers are unwilling to adopt time-consuming and laborious practices to increase their household income. Moreover, there is a positive correlation relation between cooperation membership and agents, as the cooperation may disseminate more technology information and supply related inputs to encourage adoption. Agents can relieve farmers’ concerns about product distribution after adoption. This result has important implications that related agricultural administrative departments and extension agents should emphasize to expand the channels of rice–crayfish integrated system knowledge dissemination.

### 4.2. Robustness Checks

In this section, we performed several robustness tests to further validate the stability of the results. The results are presented in Table 3. These robustness checks have confirmed the presence of the neighborhood effect on farmers’ adoption behavior.

In Column 1, we deleted farmer samples with fewer than five neighbors to eliminate the issues that farmers interact with alternative social groups or have less opportunity to interact with neighboring groups; that is, farmers with fewer than five neighbors may choose other groups to acquire agricultural information or are even unlikely to get a chance to form a community with others. After excluding the samples, the coefficient on key explanatory variable ‘neighborhood effect’ increased (from 0.379 to 0.383) and remained significantly positive.

Another concern was that we took the average adoption rate within neighboring groups in a village as the proxy variable to define the ‘neighborhood effect’. Considering rice–crayfish integrated systems are capital-intensive, farmers’ adoption behavior may be affected by neighbors having the same levels of income, instead of by the mean within neighboring group [49]. Thus, we eliminated the sample farmers in the top 30% of high-income earners in the village. The result in Column 2 reveals that the neighborhood effect was still significant.

Then, we modified the estimation model. In Column 3 of Table 3, we show the results of the ordinary least squares (OLS) estimation. The neighborhood effect remained significantly positive, which further validated the robustness of our findings.

As a final check, we followed other studies [52] in introducing external group information, which is independent from neighborhood farming groups, to construct IV variables. Then, we selected the mean adoption behavior of neighboring farmers’ relatives and friends as IV variables (The validity of the instrumental variable is verified). The average number of adoptions within relatives and friends of neighboring farmers will have an impact on neighboring farmers’ adoption behavior, but it will not affect focal farmers, which meets the requirements of instrumental variables. Furthermore, the neighbors’ relatives and friends do not live in the same village, which further suppresses the association effect. The estimations in Column 5 of Table 3 demonstrate that the neighborhood effect is significant and thus the results are confirmed.

## 5. Conclusions

Rice–crayfish integrated systems create both economic and ecological benefits. To study how this practice experienced explosive growth in China is significant for promoting the development of ecological agricultural practices in general. Farmers are subject to information inadequacies information frictions; thus, they are susceptible to their neighbors’ behavior via social interaction. In this paper, we identify the role of the neighborhood effect on farmers’ adoption behavior using 980 rural households in the middle reaches of the Yangtze River in China. To solve the potential identification problem, this paper adopts a set of empirical strategies. To control the self-selection problem, we use rural household survey data to define neighboring groups that are both spatially and socially connected. We control a series of neighboring farmers’ characteristics and village characteristics to eliminate the contextual and correlation effects. We apply the instrumental variables (IV) method to address the simultaneity problem. The empirical results reveal that a one-unit increase in neighbors’ adoption behavior increases the adoption probability of individual farmers by 0.367 units, which provides evidence of the significance of the neighborhood effect in farmers’ rice–crayfish integrated system adoption decisions. The four robustness tests also confirm the presence of the neighborhood effect in farmers’ adoption behavior.

Based on the above findings, this paper improves the understanding of farmers’ adoption in ecological agricultural practices in rural China. When the agricultural administrative institutions or extension agents attempt to develop relevant policies or improve farmers’ adoption behavior, they should not be confined to an economic perspective. The networking or social interaction between farmers should be fully exploited. Therefore, neighborhood effect can be seen as an effective approach to complement formal extension systems and promote the development of China’s ecological agriculture.

Finally, we reflect on the limitations of this paper. First, although this paper has addressed several challenges associated with our measurement of the neighborhood effect, we did not conduct a further investigation into effects such as the snowballing or social multiplier effects [13] given the limitations of cross-sectional data used in our study. Therefore, future studies may target the social multiplier effect using longitudinal data. Second, we have confirmed that the neighborhood effect matters in farmers’ adoption behavior, but we did not explore the mechanism of the neighborhood effect on farmers’ adoption behavior. It is highly suggested that future studies analyze how neighbors influence farmers’ adoption behaviors. Third, the adoption of farming practices is a process, and we only used a binary variable to measure adoption, which cannot reflect farmers’ dynamic adoption behaviors. Future studies may, therefore, extend this work by utilizing panel data.

## Figures and Tables

**Figure 1 ijerph-20-04399-f001:**
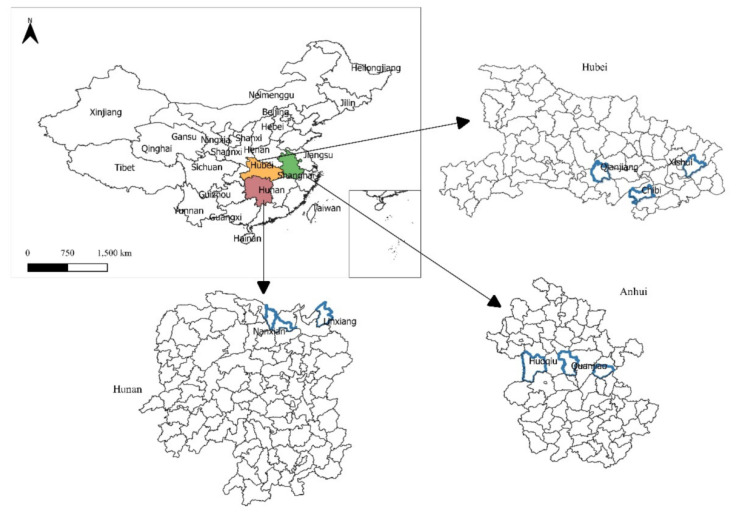
Location of the study area.

**Table 1 ijerph-20-04399-t001:** Definition of variables and summary statistics.

Variable Category	Variables	Variables Description	Mean	SD
Dependent variable	Farmers’ adoption behavior	Whether your family adopted rice–crayfish integrated systems in 2018? Dummy (1 = yes; 0 = no)	0.710	0.454
Explanatory variable	Neighborhood effect	Average adoption behavior in neighbors’ household. (range: 0–1)	0.292	0.289
Instrumental variables	Village diversity in surnames	Whether your village is a miscellaneous surname village? (1 = yes; 0 = no)	0.699	0.459
	Proportion of paddy field area	The proportion of paddy field area to cultivated land in the village. (range: 0–1)	0.848	0.141
Household characteristics	Age	Household head age. Number	54.791	9.261
	Education	Education of the household head. Number	7.276	3.204
	Risk preference ^1^	What’s your risk preference? (3 = high risk preference; 2 = neutral risk preference; 1 = low risk preference)	1.63	0.765
	Job status	Whether you engaged in part-time job? (1 = yes; 0 = no)	0.33	0.47
	Perception on economic benefits	Whether you think rice–crayfish integrated systems are highly profitable? (1 = yes; 0 = no)	0.805	0.491
	Perception on population	Rice–crayfish integrated systems are popular in your village? (5 = strongly agree; 4 = agree; 3 = not sure; 2 = disagree; 1 = strongly disagree)	3.609	0.897
	Information access	You can easily get information on rice–crayfish integrated system. (5 = strongly agree; 4 = agree; 3 = not sure; 2 = disagree; 1 = strongly disagree)	3.348	1.06
	Agricultural extension training attendance	You have attended agricultural extension training many times in 2018? (5 = frequently; 4 = often; 3 = some time; 2 = rarely; 1 = none)	3.417	1.045
	Scale of operations	How many farmlands you have operated in 2019. (mu)	91.655	202.729
	Agricultural labors	How many agricultural labors in your family? Number	2.028	0.68
	Own capital investment proportion	What’s the proportion of own possessed capital investment to the whole agricultural investment? (%)	90.099	20.949
	Cooperation membership status	Is your family any member of the village cooperation? (1 = yes; 0 = no)	0.191	0.393
	Proportion of agricultural income	What’s the proportion of agricultural income to total household income? (%)	0.693	0.272
	Plots distance	How far away is your furthest two plots? (kilometers)	0.653	1.895
Neighborhood characteristics	g_age	The average age of household heads within neighboring group. Number	54.791	4.421
	g_education	The average education of household heads within neighboring group. Number	7.276	1.48
	g_job status	The average part-time job of household heads within neighboring group. Number	0.33	0.167
	g_corperation membership status	The average member of corporation of household heads within neighboring group. Number	0.191	0.189
Village characteristics	Agents	How many agents who buy rice and crayfish within the village? Number	7.297	8.038
	Effective irrigated area	What’s the proportion of effective irrigated area in villages? (%)	94.548	11.708
	Mechanical plough road	What’s the effective traffic rate of the village mechanical plough road? (%)	90.536	17.835
Region variables	Anhui	Household from Anhui province. (1 = yes; 0 = no)	0.33	0.47
	Hunan	Household from Hunan province. (1 = yes; 0 = no)	0.335	0.472
	Hubei	Household from Hubei province. (1 = yes; 0 = no)	0.334	0.472

^1^ The measurement method of ‘risk preference’ is by asking famers the following question. If there are two varieties of rice (Seed A and Seed B), their yields may vary in the following three scenarios, what would you choose? (1 jin = 0.5 kg; 1 mu = 666 m^2^) ① A.900–1100 jin/ mu, B.800–1300 jin/ mu; ② A.900–1100 jin/ mu, B.700–1600 jin/ mu; ③ A.900–1100 jin/ mu, B.600–1800 jin/ mu. If the farmer chooses A in the three scenarios, we define them as low-risk preference; if the farmer chooses B in the three scenarios, we define them as high-risk preference. Otherwise, we define them as neutral risk preference. SD denotes standard deviation; One mu is about 0.0667 hm^2^.

**Table 2 ijerph-20-04399-t002:** Neighborhood effect in farmers’ adoption behavior.

Panel A						
Variables	Model 1: Probit		Model 2: FE		Model 3: IV Probit	
	Coef.	P	Coef.	P	Coef.	P
Neighborhood adoption behavior (NE)	0.426 ***	(0.034)	0.379 ***	(0.042)	0.367 ***	(0.124)
Age	−0.003 ***	(0.001)	−0.003 ***	(0.001)	−0.003 **	(0.001)
Educ	−0.005 *	(0.003)	−0.005 *	(0.003)	−0.005 *	(0.003)
Risk preference	0.020	(0.013)	0.020	(0.013)	0.020	(0.013)
Job status	−0.050 ***	(0.019)	−0.049 ***	(0.019)	−0.050 **	(0.020)
Perception on economic benefits	0.043 **	(0.019)	0.039 **	(0.019)	0.037 *	(0.020)
Perception on population	0.030 ***	(0.010)	0.027 ***	(0.010)	0.027 **	(0.012)
Information access	0.075 ***	(0.011)	0.077 ***	(0.011)	0.077 ***	(0.015)
Extension training attendance	−0.011	(0.009)	−0.012	(0.009)	−0.012	(0.010)
Scale of operations	−0.000	(0.000)	−0.000	(0.000)	−0.000	(0.000)
Agricultural labors	−0.009	(0.013)	−0.010	(0.013)	−0.010	(0.014)
Investment proportion	−0.002 ***	(0.001)	−0.002 ***	(0.001)	−0.002 **	(0.001)
Cooperation membership status	0.076 ***	(0.029)	0.072 **	(0.029)	0.072 ***	(0.027)
Proportion of agricultural income	−0.084 **	(0.037)	−0.089 **	(0.037)	−0.089 **	(0.042)
Plots distance	−0.008 **	(0.003)	−0.007 **	(0.003)	−0.008	(0.005)
g_age	0.003	(0.003)	0.001	(0.003)	0.001	(0.006)
g_educ	0.005	(0.008)	0.003	(0.009)	0.003	(0.009)
g_Job status	0.063	(0.053)	0.051	(0.053)	0.047	(0.062)
g_ corperation Membership status	−0.103 *	(0.053)	−0.097 *	(0.053)	−0.096	(0.065)
Agents	0.004 **	(0.001)	0.004 ***	(0.002)	0.004	(0.003)
Effective_irrigated_area	−0.000	(0.001)	−0.000	(0.001)	0.000	(0.001)
Mechanical_plough_road	0.001	(0.001)	0.001	(0.001)	0.001	(0.001)
Hubei			0.056 **	(0.027)	0.052	(0.044)
Anhui			0.002	(0.026)	−0.006	(0.027)
Panel B: First-stage estimation results						
Village diversity in surnames					−0.011 ***	(0.001)
Proportion of paddy field area					0.027 ***	(0.004)
First-stage F value—Weak identification test					61.23	
DWH *p*-Value—Endogeneity test					0.090	
Amemiya-Lee-Newey minimum chi-sq statistic *p*-Value—Over-identification test					0.632	

***, **, * denotes *p* < 0.01, *p* < 0.05, and *p* < 0.1, respectively. The ‘Coef.’ presented in Panel A is the marginal effects (dy/dx) of the variables, taking Hunan as the reference group.

**Table 3 ijerph-20-04399-t003:** Robustness checks of the neighborhood effect on farmers’ adoption behavior.

	Robustness Checks 1	Robustness Checks 2	Robustness Checks 4	Robustness Checks 5
	coef. (*p*-Value)	coef. (*p*-Value)	coef. (*p*-Value)	coef. (*p*-Value)
Neighborhood effect	0.383 *** (0.122)	0.413 *** (0.128)	0.803 *** (0.148)	0.372 *** (0.100)
Instrumental variables	YES	YES	YES	YES
Household characteristics	Controlled	Controlled	Controlled	Controlled
Neighborhood characteristics	Controlled	Controlled	Controlled	Controlled
Village characteristics	Controlled	Controlled	Controlled	Controlled
Provincial dummies	Controlled	Controlled	Controlled	Controlled
Observations	930	727	980	980

*** denotes *p* < 0.01.

## Data Availability

Not applicable.

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
