# Peer review of "How Neighbors Influence Rice–Crayfish Integrated System Adoption: Evidence from 980 Farmers in the Lower and Middle Reaches of the Yangtze River"

_ijerph, 2023, doi:10.3390/ijerph20054399_

Round 1
Reviewer 1 Report
The paper aims to demonstrate whether neighbors would influence farmers’ adoption behavior on rice-crayfish integrated system in the area of lower and middle reaches of Yangtze River in China.
My congratulations to the authors. The paper is very interesting, original and well structured. It would almost be ready for publication, if it were improved in the use of English (e.g. verb tenses, conjunction verbs with third person singular/plural).
The section dedicated to “Empirical results” can be improved with small changes, for instance by moving the lines 302-315 to the end of the paragraph and avoiding redundancy (e.g. lines 261-263 and lines 296-297; lines 267-273 and lines 330-337).
I highly recommend presenting Discussion as a single section separated from Conclusions, considering the instruction for authors by the IJERPH (Authors should discuss the results and how they can be interpreted in perspective of previous studies and of the working hypotheses. The findings and their implications should be discussed in the broadest context possible and limitations of the work highlighted. Future research directions may also be mentioned. This section may be combined with Results.).
Finally, I would like to know why in China there is information friction between farmers and extension agents (line 74) and why “the higher proportion of own possessed capital investment to the whole agricultural investment, the less probability of farmers’ adoption” (lines 344-346) .
Other secondary suggestions:
- Page 7, line 258: I would suggest “As mentioned above” as incipit.
- I also suggest resorting transitional phrases in order to tie sentences together (e.g. “it must be clarified that” instead of “Firstly” or “Second”).
- Please, add Hunan Province data in Table 2 and check numbers 3 and 4 in the last columns of Table 3, and in lines 382-383 as well.
Reviewer 2 Report
The article “How neighbors influence ecological agricultural farming adoption: evidence from 980 farmers in the lower and middle reaches of Yangtze River” discussed whether neighbors would influence farmers’ adoption behavior on the rice-crayfish integrated system is an interesting topic that is worthy of study. For your reference, here are some suggestions:
Lines 53-55. “Moreover, integrated system requires large investments (e.g., proprietary equipment) and more flood-and-drought-associated risks (Bashir et al, 2020), which may expose farmers to increased risk levels.” How to understand that integrated system requires more flood-and-drought-associated risks?
Data: The sample size is inconsistent. As follows, Title: “evidence from 980 farmers in the lower and middle reaches of Yangtze River” & Lines 183-184. “We have a final sample of 851 households in the sample.” & Lines 204-205. “We collected 980 farmer households’ samples in three major provinces in the lower and middle reaches of the Yangtze River in China in 2019”.
Lines 207-208. “We define farmers living in the same administrative village as neighboring group.” Why administrative village rather than a natural village is used? As known, a natural village is one where the inhabitants naturally developed it as a result of their long-term residence, and an administrative village may have a relatively wide region under its control.
Table 1: The mean and standard deviation of the dependent variable are missing.
Line 279: Why fixed effect (FE) is employed?
Results and Discussion. According to the author guidelines of IJERPH, the section of Results should provide a concise and precise description of the experimental results, their interpretation, as well as the experimental conclusions that can be drawn. The section of Discussion should discuss the results and how they can be interpreted from the perspective of previous studies and of the working hypotheses. The findings and their implications should be discussed in the broadest context possible. Future research directions may also be highlighted.
Therefore, parts of the results can be transferred to the discussion, that is, the description of the model findings and their interpretation is provided in this section of Results, while the discussion of the results and how they can be interpreted from the perspective of previous studies is provided in this section of Discussion.
Reference: In the text, reference numbers should be placed in square brackets [ ] and placed before the punctuation; for example [1], [1–3] or [1,3]. For embedded citations in the text with pagination, use both parentheses and brackets to indicate the reference number and page numbers; for example [5] (p. 10), or [6] (pp. 101–105).
Author 1, A.B.; Author 2, C.D. Title of the article. Abbreviated Journal Name Year, Volume, page range.
Reviewer 3 Report
This paper introduces an important topic for environmental research studies and fits well in the general purpose of the journal. Following the abstract, the objective of the paper is to identify whether neighbors would influence farmers’ adoption behavior on rice-crayfish integrate system’.
Despite of the focus of the general purpose of the article, this proposal has several weaknesses: some concepts are not clearly defined, the abstract should be improved, the literature review requires fine tuning, the methods should be detailed and improved, external validity was not implemented, and the conclusion should be completed and reframed…
The title of the paper suggests it is quite ambitious as it intends to provide an examination of the ‘how neighbors influence ecological agricultural farming adoption’.
The abstract should be rewritten and streamlined. The theoretical/conceptual approaches are not disclosed. Instead of focusing on the main hypothesis and the conceptual approaches, the abstract details the use of a ‘instrumental variable method’ and ‘robustness check’. Why not relevance is given to the conceptual approach ? Is it less important ?
The use of a ‘representative sample’ and ‘after we conducted multiple robustness checks’ should be removed. We expect the authors will that that into consideration (redundancy). Such statements are not relevant into an abstract.
The structure of the paper also should be improved. The paper is unbalanced.
The General Introduction occupies 2.5 pages and the General Conclusion only approximately 0.5 pages. This is not appropriate.
Further, after the general introduction the authors move directly to the data and methods (4 pages). There should be an intermediate section developing and presenting the literature review. A part of the literature review is developed in the general introduction. A specific section should be dedicated to the literature review and the paper should be balanced.
The literature review should present a table with the main authors, variables to assess the importance of the determinants of adoption of such practices, and the influence of neighbors, findings, etc. The two points indicated in as an extension of the literature review (lines 121 and following) should be derived from the table.
Some related and relevant theories should be integrated. The Theory of the Planned Behavior and recent advances in Nudge Theory should be discussed. The literature review should position your work with respect to such conceptual developments in farmers’ behavior and adoption of new farming practices.
The data and methods make reference to a questionnaire. Very little information is available about the how the questionnaire was built and administrated. How did you build and test the questionnaire ? who administrated and collected the data in the questionnaire (self-completed survey ?) ? what is the structure of the questionnaire ?
The main goal of the questionnaire should be stated in the introduction and not repeated across the body of the article (see lines 224 and 225).
The Adoption of farming practices is a process. Using a binary variable to measure the adoption if farmers’ practices is a strong limitation of your research (see equation, line 228). This should be reported in the conclusions.
Some questions included in the questionnaire use poor measurements. A very good example is the measurement of ‘risk preference’. Asking directly people what is their risk preference is a wrong approach. Risk is often measured in a multi-dimension scale in marketing or business studies and agricultural studies (see, for example, Sarma, 2022 on The Theory of the Planned Behavior). A survey of the literature on perceived risks was realized by Iyer et al.(2020) in Europe and Jin et al.(2017) in China. This literature needs to be taken into consideration in the definition of the constructs.
The general conclusion should start by presenting the problem statement and stating the main findings. It should detail the limitations: theoretical, practical, and methodological. Many limitations related to the conceptualization (theory of planned behavior, nudge theory….), methods (construction of the questionnaire, incentives provided to the respondents, administration of the questionnaire, variable definition…) were not taken into consideration.
